
# Fragmentation inside PTR-based mass spectrometers limits the detection of ROOR and ROOH peroxides

Haiyan Li[1,2,†], Thomas Golin Almeida[3,†], Yuanyuan Luo[2], Jian Zhao[2], Brett B. Palm[4], Christopher D. Daub[3], Wei Huang[2], Claudia Mohr[5], Jordan E. Krechmer[6], Theo Kurtén[3], Mikael Ehn[2]

[1]School of Civil and Environmental Engineering, Harbin Institute of Technology, Shenzhen, 518055
[2]Institute for Atmospheric and Earth System Research / Physics, Faculty of Science, University of Helsinki, Helsinki, 00014, Finland
[3]Department of Chemistry, University of Helsinki, Helsinki, 00014, Finland
[4]Department of Atmospheric Science, University of Washington Seattle, Washington, WA, 98195, USA
[5]Department of Environmental Science, Stockholm University, 11418, Stockholm, Sweden
[6]Aerodyne Research, Inc., Billerica, MA, 01821, USA
†These authors contributed equally to this work.

*Correspondence to*: Haiyan Li (lihaiyan2021@hit.edu.cn), Mikael Ehn (mikael.ehn@helsinki.fi), Theo Kurtén (theo.kurten@helsinki.fi)

**Abstract.** Proton-transfer-reaction (PTR) is a commonly applied ionization technique for mass spectrometers, where hydronium ions ($H_3O^+$) transfer a proton to analytes with higher proton affinities than the water molecule. This method has most commonly been used to quantify volatile hydrocarbons, but later generation PTR-instruments have been designed for better throughput of less volatile species, allowing detection of more functionalized molecules as well. For example, the recently developed Vocus PTR time-of-flight mass spectrometer (PTR-TOF) has been shown to agree well with an iodide

adduct based chemical ionization mass spectrometer (CIMS) for products with 3-5 O-atoms from oxidation of monoterpenes ($C_{10}H_{16}$). However, while several different types of CIMS instruments (including those using iodide) detect abundant signals also at "dimeric" species, believed to be primarily ROOR peroxides, no such signals have been observed in the Vocus PTR, even though these compounds fulfil the condition of having higher proton affinity than water.

More traditional PTR instruments have been limited to volatile molecules as the inlets have not been designed for transmission

of easily condensable species. Some newer instruments, like the Vocus PTR, have overcome this limitation, but are still not able to detect the full range of functionalized products, suggesting that other limitations need to be considered. One such limitation, well-documented in PTR literature, is the tendency of protonation to lead to fragmentation of some analytes. In this work, we evaluate the potential for PTR to detect dimers and the most oxygenated compounds, as these have been shown to be crucial for forming atmospheric aerosol particles. We studied the detection of dimers using a Vocus PTR-TOF in laboratory

experiments as well as through quantum chemical calculations. Only noisy signals of potential dimers were observed during experiments on the ozonolysis of the monoterpene α-pinene, while a few small signals of dimeric compounds were detected during the ozonolysis of cyclohexene. During the latter experiments, we also tested varying the pressures and electric fields in the ionization region of the Vocus PTR-TOF, finding that only small improvements were possible in the relative dimer contributions. Calculations for model ROOR and ROOH systems showed that most of these peroxides should fragment



partially following protonation. With inclusion of additional energy from the ion-molecule collisions driven by the electric fields in the ionization source, computational results suggest substantial or nearly complete fragmentation of dimers. Our study thus suggests that while the improved versions of PTR-based mass spectrometers are very powerful tools for measuring hydrocarbons and their moderately oxidized products, other types of CIMS are likely more suitable for the detection of ROOR and ROOH species.

## 1 Introduction

Volatile organic compounds (VOCs) are emitted into the atmosphere from a variety of sources, both biogenic (Kesselmeier and Staudt, 1999;Sindelarova et al., 2014) and anthropogenic (Friedrich and Obermeier, 1999;Theloke and Friedrich, 2007;Huang et al., 2011). The major atmospheric loss process of these VOCs is oxidation, and the oxidation process can contribute to the formation of harmful pollutants such as tropospheric ozone (Carter, 1994;Ying and Krishnan, 2010) and
organic particulate matter (Chen et al., 2011;Ehn et al., 2014). The ability of a specific VOC to produce organic aerosol depends on how efficiently the VOC is converted to low-volatile species during the oxidation process (Donahue et al., 2012). Oligomerization and oxygenation tend to lower the volatility, making the products more likely to condense into the particulate phase, while fragmentation typically has the opposite effect by increasing the volatilities of the products (Jimenez et al., 2009). As atmospheric aerosol particles greatly affect Earth's radiative balance (Charlson et al., 1992;Rap et al., 2013), the relative
branching ratios between these three pathways, and the oxidation process in general, is of great importance.

In the last decade, online chemical ionization mass spectrometers (CIMS) have become increasingly applied in atmospheric studies of VOC oxidation (Bertram et al., 2011;Jokinen et al., 2012;Lee et al., 2014b;Krechmer et al., 2018;Lopez-Hilfiker et al., 2019). Their high sensitivity and high time resolution have provided detailed information on a vast number of oxidation products, both in gas and aerosol phase, leading to a suite of new insights and breakthroughs (Ehn et al., 2014;Bianchi et al.,
2019;Mohr et al., 2019). In some cases, nearly full mass closure has been achieved between precursors and products (Hansel et al., 2018;Isaacman-VanWertz et al., 2018). Ultimately, the type of CIMS being deployed, defined primarily by the choice of reagent ion and the inlet design, determine the selectivity of the instrument (Riva et al., 2019). Some instruments are mainly selective towards the most oxygenated species, like the Chemical Ionization Atmospheric Pressure interface Time-of-Flight mass spectrometers (CI-APi-TOF) using $NO_3^-$ as the reagent ion (Hyttinen et al., 2015;Yan et al., 2016), while others primarily
detect moderately oxidized products, like the iodide ($I^-$)-CIMS (Lee et al., 2014a). Many of these commonly used selective ion-molecule reactions have been solidly supported and explained through computational studies of clustering strengths between reagent ions and different types of organic compounds (Kupiainen-Määttä et al., 2013;Iyer et al., 2017;Hyttinen et al., 2018). For the detection of precursor VOCs, the most typical CIMS type used has been proton-transfer-reaction (PTR) mass spectrometry, which is based on the transfer of a proton from a hydronium ion ($H_3O^+$) to an analyte with higher proton
affinity (Yuan et al., 2017). Earlier versions of PTR instruments have been designed to measure only very volatile compounds,



but more recent designs like the PTR3 (Breitenlechner et al., 2017) and the Vocus PTR (Krechmer et al., 2018) have shown their ability to detect also less volatile oxidation products in addition to the precursor VOC.

Good agreement between different CIMS types has been seen for specific compound groups. As shown recently e.g. in a CIMS intercomparison study (Riva et al., 2019), moderately oxidized monomeric products agreed well between an iodide CIMS and

a Vocus PTR, while highly oxygenated accretion products agreed well between CI-APi-TOF instruments using either $NO_3^-$ or $C_4H_{12}N^+$ (butylamine) as reagent ions. These accretion products are believed to be primarily peroxide dimers, ROOR, formed from cross reactions of peroxy radicals, $RO_2$ (Schervish and Donahue, 2020;Tomaz et al., 2021). Similarly, aminium CI-APi-TOF and the ammonium adduct PTR3 matched well over a wide range of oxidation products under very clean conditions (Berndt et al., 2018b). However, to our knowledge, with the exception of the PTR3, temporal agreement between any CIMS

instruments (including the iodide CIMS and the Vocus PTR) and a nitrate CI-APi-TOF has never been shown for the most highly oxygenated products. Additionally, no accretion products, i.e. "dimers" (Berndt et al., 2018b;Valiev et al., 2019), have been reported for any other PTR instruments than the PTR3 (where ionization can also take place as ligand switching from protonated water clusters) (Breitenlechner et al., 2017), although the other chemical reagent ions regularly detected them. As one of the very few types of commonly used CIMS which has also been optimized for measurements of hydrocarbon VOC,

these apparent limitations are disappointing as PTR instruments like the Vocus PTR could otherwise provide a uniquely broad range of observations of both precursors and products. One possible reason for this limitation, also speculated by Riva et al. (2019) regarding the lack of detected dimer species by a Vocus PTR, is that protonation leads to fragmentation, a process which is known to be common for different types of hydrocarbons in PTR instruments (Tani et al., 2003;Aprea et al., 2007;Gueneron et al., 2015)..

In this work, we aimed to characterize the ability of a PTR mass spectrometer to detect ROOR and ROOH species, both experimentally and computationally. We performed experiments using a Vocus PTR-TOF to measure oxidation products from different precursors, and varied the settings under which the ionization took place, to identify parameters that influence the detection of dimeric compounds. Quantum chemical calculations were performed for both ROOR and ROOH systems to estimate to what extent, and through which channels, the protonation and subsequent ion transport could lead to fragmentation

before detection.

## 2 Materials and methods

### 2.1 Experimental methods

To test the ability of PTR mass spectrometry to detect ROOR species, we deployed a Vocus PTR-TOF in chamber experiments of VOC oxidation known to produce a wide range of products with different functionalities. We acknowledge that the Vocus

PTR is only one of many PTR designs, and while the basic principle of proton transfer from hydronium ions is common for all instruments, differences in design and operation may impact the general applicability of our experimental findings. In the chamber experiments, ozone was reacted with either α-pinene, a monoterpene and one of the most abundantly emitted VOCs



globally (Guenther et al., 2012), or cyclohexene, a commonly used surrogate for monoterpenes. These systems were chosen because they are known to produce a large number of different dimer species (Berndt et al., 2018a;Rissanen et al., 2014;Hansel

et al., 2018).

The laboratory experiment was conducted in a 2 m$^3$ atmospheric simulation Teflon FEP (fluorinated ethylene propylene) chamber at the University of Helsinki. More details about the chamber have been presented by Riva et al. (2019) and Peräkylä et al. (2020). The chamber was operated under steady-state conditions with a constant flow of reactants and oxidants continuously added to the chamber. The chamber was maintained at a slight overpressure, and the average residence time was

~45 min. The majority of the chamber flow was sampled by a variety of instruments, including the Vocus PTR-TOF, with the rest flushed into an exhaust line. We performed the α-pinene/cyclohexene ozonolysis experiments in the absence of an OH scavenger at room temperature (27 ± 2 °C) and under dry conditions (RH < 1%). The dry air was purified with a clean air generator (AADCO model 737-14, Ohio, USA) and injected into the chamber along with the gaseous reactants. Ozone was generated by injecting purified air through an ozone generator (Dasibi 1008-PC). α-pinene or cyclohexene addition into the

chamber was controlled by a heated syringe pump. Ozone concentrations in the chamber were monitored by a UV photometric analyzer (Model 49P, Thermo-Environmental) during cyclohexene ozonolysis, but the analyser was not available during α-pinene ozonolysis experiments.

The Vocus PTR-TOF utilizes a low-pressure discharge reagent-ion source to produce $H_3O^+$ reagent ions, which are subsequently mixed with sample air in a focusing ion-molecule reactor (FIMR) (Krechmer et al., 2018). The FIMR consists

of a resistive glass tube with four quadrupole rods mounted radially on the outside. The glass tube improves measurement delay times and transmission by eliminating any metal surfaces for gaseous molecules or ions to adsorb to. The ions are transported through the 10 cm-long FIMR using a DC (direct current) electric field, with a typical potential gradient of 450-650 V (500 V in these experiments as default) and a variable quadrupole RF (radio frequency) field applied transversely to focus ions along the central axis. The pressure in the FIMR was kept at 1.4 mbar, except for experiments where the influence

of pressure on the mass spectra was probed. In this study, sample air with a flow rate of 4.5 L min$^{-1}$ was drawn through 1 m long PTFE (polytetrafluoroethylene) tubing (6 mm o.d., 4 mm i.d.), out of which around 0.1 L min$^{-1}$ went into the FIMR and the remainder was directed to the exhaust. The Vocus ionization source was operated with a 15 sccm flow of pure water vapor. For a molecule R with higher proton affinity than $H_2O$, ionization can occur via proton transfer from $H_3O^+$ to produce the product ion ($[RH]^+$)

$$R + H_3O^+ \rightarrow [RH]^+ + H_2O \tag{Rp}$$

The PTR technique is considered a soft ionization method when compared to, e.g., electron impact ionization. However, the $[RH]^+$ ions produced from the above reaction can still undergo fragmentation in the reactor due to the added energy from the exothermic protonation reaction as well as collisions (Gueneron et al., 2015;Tani et al., 2003;Tani et al., 2004). While higher collision energy reduces clustering of the reagent and product ions with water, it can also promote fragmentation of the

protonated ions. The fragmentation inside the PTR strongly depends on the parameter E/N, the ratio of the guiding electric



field to the number density of the gas in the PTR drift tube. An increase in E/N results in more energetic collisions and thus an increased degree of fragmentation (Gueneron et al., 2015).

## 2.2 Computational methods

To investigate fragmentation patterns in a less instrument-specific manner, quantum chemical methods were used to calculate

the energetics of gas-phase fragmentation pathways available to a selection of protonated model ROOR and ROOH compounds. In an elementary reaction, reactants and products are connected by a transition state, defined as the point of highest potential energy in the reaction coordinate. The transition state represents a barrier that must be surpassed for the reaction to occur, and knowledge of its energy relative to the reactant (i.e. the barrier height) allows one to predict the thermal rate of the reaction. However, the studied reactions happen in low pressures and the preceding protonation step is exothermic. Collisions

with bath gas molecules may be too rare to rapidly dissipate the energy accumulated during protonation, and a portion of the peroxide molecules may fragment at non-thermal rates. Thus, appropriate analysis of fragmentation in this case requires treatment of collisional energy transfer and energy-specific microcanonical rates, which are included in master equation solvers. Calculation of the Potential Energy Surface (PES), i.e. the energy of each reactant, intermediate, transition state and product involved in a reaction mechanism, was employed to predict how fast a series of model protonated peroxides fragment

and which would be their product yields.

The model ROOR systems studied were: methyl peroxide (MeOOMe), ethyl peroxide (EtOOEt), isopropyl peroxide (iPrOOiPr), hydroxyethyl methyl peroxide (HOEtOOMe), oxoethyl methyl peroxide (O=EtOOMe) and acetyl peroxide (AcOOAc); and the model ROOH systems were: methyl hydroperoxide (MeOOH), ethyl hydroperoxide (EtOOH), isopropyl hydroperoxide (iPrOOH), tert-butyl hydroperoxide (tButOOH), hydroxyethyl hydroperoxide (HOEtOOH), oxoethyl

hydroperoxide (O=EtOOH), peracetic acid (AcOOH) and performic acid (OCHOOH). In addition, selected ROOR and ROOH systems derived from cyclohexene oxidation initiated by OH radical and ozone were also studied (Figure 1). These compounds are 2,2'-peroxy bis(hexanedial) (**A**), 2-hydroperoxyhexanedial (**B**), 2,2'-peroxy bis(cyclohexanol) (**C**) and 2-hydroperoxycyclohexanol (**D**).

**Figure 1. Investigated ROOR and ROOH species derived from oxidation of cyclohexene by OH radical and O₃.**





Initial systematic conformer search was performed for each species involved in the studied reactions with the Merck Molecular Force Field (MMFF) method (Halgren, 1999) implemented in Spartan '18 (2018). All conformers were then optimized at the B3LYP/6-311++G** level of theory (Becke, 1993;Lee et al., 1988), and the ones yielding the lowest electronic energies (within 2 kcal.mol$^{-1}$) were reoptimized at the ωB97XD/aug-cc-pVTZ level (Chai and Head-Gordon, 2008;Kendall et al., 1992), using Gaussian'16 RevC.01 (Frisch et al., 2016). Subsequent frequency calculations provided vibrational frequencies and rotational constants, and served to ensure that all optimized geometries correspond to stationary points in the PES. Conformer geometries with similar electronic energy and electric dipole (difference less than 10-5 Hartree and 0.015 Debye respectively) were identified as duplicates and removed (Møller et al., 2016). Initial guesses for some of the transition state (TS) geometries of the simplest systems, MeOOMe and MeOOH, were taken from a study published by Schalley et al. (1997). Transition states for other steps were found by doing relaxed scans over critical bond lengths or angles, and some were found during the geometry optimization of conformers of unrelated transition states. Intrinsic Reaction Coordinate (IRC) calculations were done for each transition state, at the B3LYP/6-311++G** level of theory, in order to check if they correctly connect to the assumed reactant and product minima. Finally, more accurate electronic energies were obtained for the lowest energy conformer of each species, by performing single point RHF-RCCSD(T)-F12a/VDZ-F12 calculations (Knizia et al., 2009), with the Molpro 2019.2 software (Werner et al., 2012).

Once the PES of each model ROOR and ROOH system was obtained, the time-dependent species distribution profile was estimated with MESMER (Master Equation Solver for Multi-Energy Well Reactions) version 5.2 (Glowacki et al., 2012). Rice-Ramsperger-Kassel-Marcus (RRKM) theory was used to treat isomerization reaction steps and the reverse Inverse Laplace Transformation (ILT) method was used for irreversible dissociation steps. The initial charged species (RH+) were modelled as having emerged from a barrierless dissociation reaction, $A(E_{ex}) \rightarrow RH^+ + H_2O$, where $E_{ex}$ is the excess energy in reactant A, which was assumed to be equal to the zero-point corrected energy of reaction Rp (ΔEzp). The exponential down model was employed to treat collisional energy transfer. Lennard-Jones parameters and $\langle \Delta E_{down} \rangle$ values used were fitted to molecular dynamics simulation results for thermalization rates of three model (ROOR)H+ species. Further details on the methods are found in the Supplementary Information.

## 3 Results and discussion

### 3.1 Oxidation products from α-pinene ozonolysis

Figure 2a shows the average mass spectra of Vocus PTR-TOF measurements during α-pinene ozonolysis ([α-pinene] = 82.4 ± 148.8 ppb), covering both α-pinene and α-pinene oxidation products. Consistent with previous studies, no α-pinene dimers were observed by Vocus PTR-TOF at atmospherically relevant α-pinene concentrations (Riva et al., 2019;Li et al., 2020;Li et al., 2021), despite many other types of CIMS routinely detecting abundant dimer formation under similar conditions (Zhao et al., 2018;Riva et al., 2019). To explore potential sensitivity limitations, we increased the α-pinene concentration to as high as



1 ppm (signal around $2.0 \times 10^6$ cps). As shown in the mass spectra in Fig. 2a, only four different oxygen-containing ions with carbon number around 20 were observed, although at extremely low concentrations, with the molecular formulae of $C_{18}H_{30}O_2$,

$C_{20}H_{30}O_1$, $C_{20}H_{30}O_2$, and $C_{20}H_{32}O_3$. It is hard to speculate how such compositions would arise from the ozonolysis or OH oxidation of the precursor molecule without additional in-source reactions. The production of $C_{20}H_{32}H^+$, which was also observed, was most probably caused by the secondary association reactions between protonated α-pinene with α-pinene in the FIMR. A previous study by Bernhammer et al. (2018) has observed unexpected ion signals of $C_{10}H_{16}H^+$ and $C_{20}H_{32}H^+$ during pure isoprene oxidation. Similar to the results of Bernhammer et al. (2018), the signals of $C_{20}H_{32}H^+$ show a quadratic

dependency on the signals of $C_{10}H_{16}H^+$ in our study (Fig. S1).

During the experiment, α-pinene signals varied from $0.5 \times 10^6$ cps to $2.0 \times 10^6$ cps and then to $0.2 \times 10^6$ cps due to changing the injection (Fig.2b). Correspondingly, significant variations were observed for the $C_{10}$ monomers from α-pinene oxidation (i.e., $C_{10}H_{16}O_2$, $C_{10}H_{16}O_3$, $C_{10}H_{14}O_5$, and $C_{10}H_{14}O_6$). However, for the observed oxygenated compounds with carbon number around 20 (i.e., $C_{18}H_{30}O_2$ and $C_{20}H_{30}O_2$), only noisy signals were detected even when α-pinene concentrations reached 1 ppm (i.e.,

signal $\sim 2.0 \times 10^6$ cps). It is possible that all the signals observed in the dimer range are simply clusters formed in the FIMR, similar to $C_{20}H_{32}H^+$. We also emphasize that the poor detection of signals in the region of potential α-pinene dimers in the Vocus PTR-TOF is not merely caused by a low transmission of ions with larger masses. This is for example seen from data from a subsequent study using the same chamber and Vocus PTR-TOF, where the diterpene kaurene ($C_{20}H_{32}$) was oxidized, and we saw significant signals of $C_{19}$ and $C_{20}$ oxidation products (i.e., $C_{19}H_{30}O$ and $C_{20}H_{32}O_3$) from Vocus PTR-TOF

measurements (Fig. S2). If the lack of dimers is due to fragmentation, these fragments will most likely overlap with larger signals from monomers, and therefore the very minor contribution of potential dimers to the mass spectrum remains the main experimental result. Similarly, since only the elemental composition of the ions is known, we cannot draw any conclusions on the fates of ROOH species, except to say that we did not detect any signals from typical HOMs as commonly observed by the nitrate CI-APi-TOF.

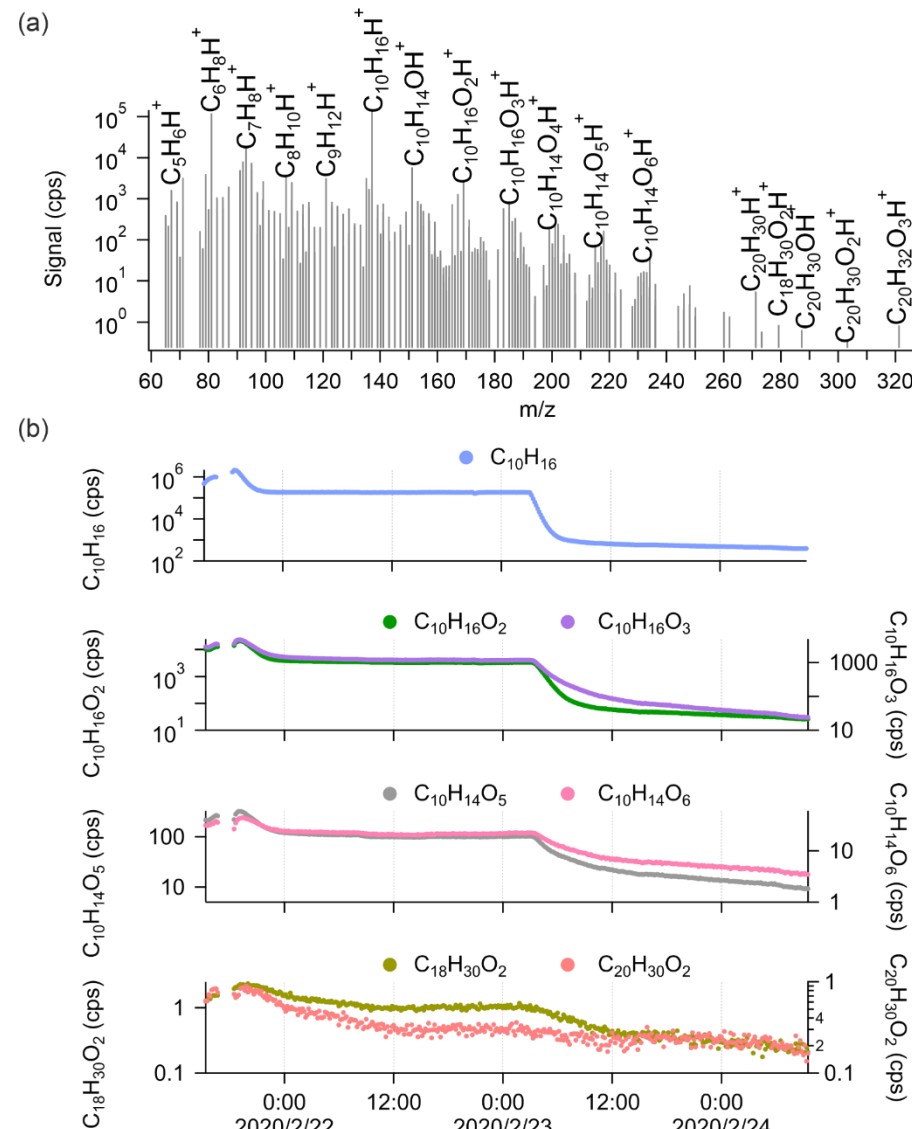

**Figure 2. (a) The average mass spectrum of Vocus PTR-TOF measurements during α-pinene (average concentration of 82.4 ± 148.8 ppb) ozonolysis in log scale. The peaks of α-pinene, its major fragments inside the Vocus PTR-TOF, and some α-pinene oxidation products are labelled in the spectrum. (b) Time series of α-pinene ($C_{10}H_{16}$) and its oxidation products during the experiment, including α-pinene monomers ($C_{10}H_{16}O_2$, $C_{10}H_{16}O_3$, $C_{10}H_{14}O_5$, and $C_{10}H_{14}O_6$) and potential α-pinene dimers ($C_{18}H_{30}O_2$ and $C_{20}H_{30}O_2$). α-pinene signals were varied by changing the injection.**

### 3.2 Oxidation products from cyclohexene ozonolysis

We further performed cyclohexene ($C_6H_{10}$) ozonolysis experiments ([cyclohexene] = 18.7 ± 13.1 ppb; [$O_3$]= 26.4 ± 9.3 ppb) in the chamber to explore the detection of dimers by Vocus PTR-TOF with a simpler system. From the average mass spectrum in Fig. 3a, $C_5H_6H^+$ and $C_6H_8H^+$ are the most abundant fragments of cyclohexene inside the Vocus PTR-TOF. The variations of $C_5H_6H^+$ and $C_6H_8H^+$ correlated closely with those of cyclohexene during the experiment. Cyclohexene oxidation products





with varying oxidation degrees were observed, with the number of oxygen atoms ranging from one to six (Fig. 3a). More oxygenated compounds with nO>6 have previously been seen from cyclohexene ozonolysis using iodide, nitrate, and $NH_4^+$ CIMS (Iyer et al., 2017;Hansel et al., 2018;Rissanen et al., 2014).

Only two potential cyclohexene dimers were clearly detected by Vocus PTR-TOF, $C_{12}H_{20}O_4$ and $C_{12}H_{22}O_4$. The signal
intensities were clearly lower than those of the most abundant monomeric species, but the difference was not as large as in the case of α-pinene. As displayed in Fig. 3b, the $C_{12}H_{22}O_4$ signal was around seven times higher than that of $C_{12}H_{20}O_4$, and considering the standard dimer-forming reaction, $RO_2 + RO_2 \rightarrow ROOR + O_2$, $C_{12}H_{22}O_4$ can be formed by the reaction of two $C_6H_{11}O_3$ peroxy radicals. The $C_6H_{11}O_3$ radical has been previously detected as a major $RO_2$ radical from cyclohexene ozonolysis in the absence of an OH scavenger (Iyer et al., 2017;Hansel et al., 2018). It is the main expected OH-derived $RO_2$,
following OH addition to the double bond and subsequent addition of $O_2$. Ozone-derived $RO_2$ would contain fewer H-atoms, and should be more abundant in this system, but no such dimeric compounds were observed. The time series of $C_{12}H_{20}O_4$ and $C_{12}H_{22}O_4$ showed high similarities to those of monomeric oxidation products, with correlation coefficients ($r^2$) of mostly larger than 0.9, which is higher than their correlation coefficient with cyclohexene itself. As in the case of α-pinene, we cannot separate any potential dimer fragments from the mass spectra, since all the oxidation products have very high correlations with
each other. We can also note that the lack of dimers cannot be attributed to low transmission in the dimer range, as the cyclohexene dimers ($C_{12}$) are similar in mass as α-pinene monomers ($C_{10}$).

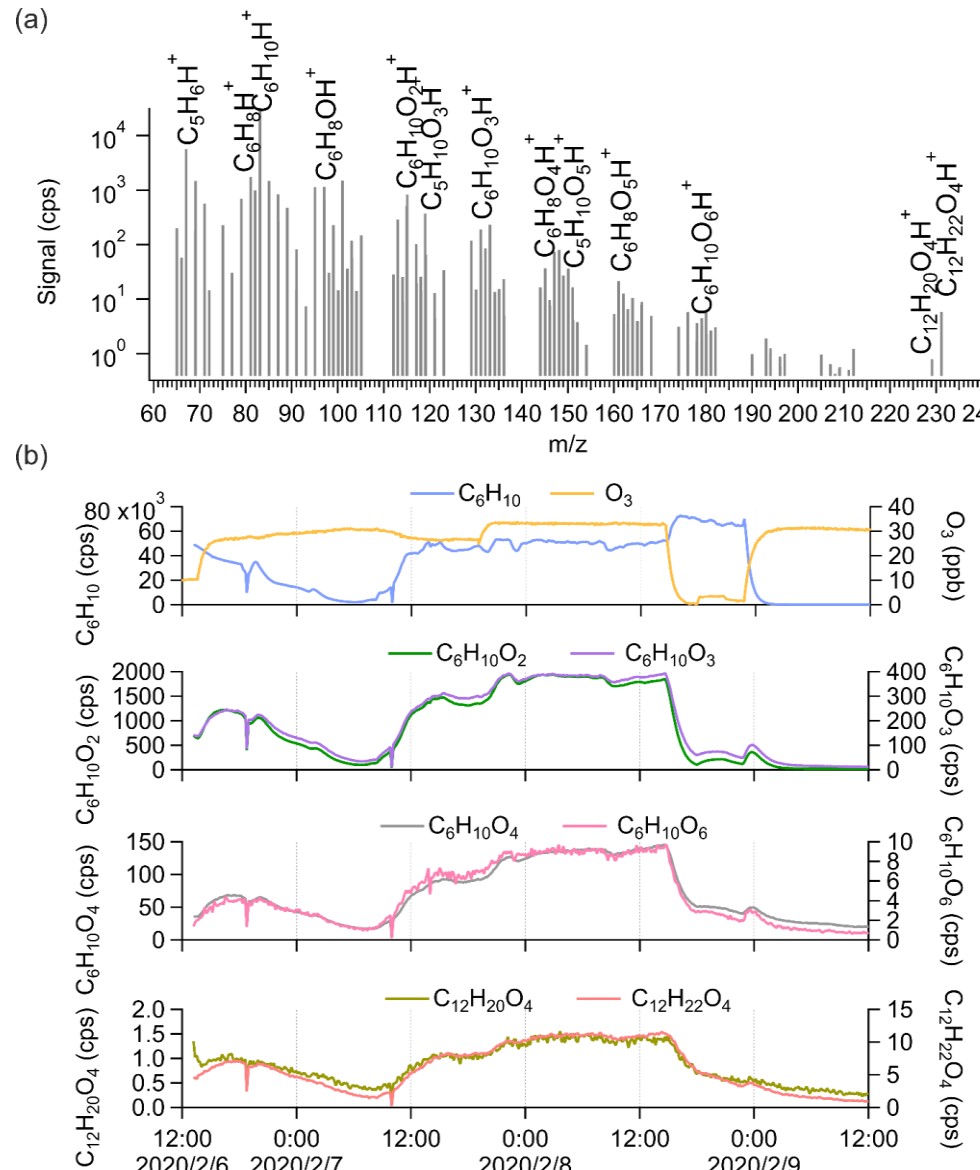

**Figure 3. (a) The average mass spectrum of Vocus PTR-TOF measurements from cyclohexene ozonolysis. The peaks of cyclohexene, its major fragments inside the Vocus PTR-TOF, and some cyclohexene oxidation products are labelled. (b) Time series of cyclohexene ($C_6H_{10}$), $O_3$, cyclohexene monomers ($C_6H_{10}O_2$, $C_6H_{10}O_3$, $C_6H_{10}O_4$, and $C_6H_{10}O_6$), and potential cyclohexene dimers ($C_{12}H_{20}O_4$ and $C_{12}H_{22}O_4$) during the experiment. Cyclohexene signals were varied by changing the injection.**

### 3.3 Influence of ionization settings on dimer detection

To further get insights into the potential fragmentation of dimers inside the Vocus PTR-TOF, we varied the FIMR pressure and DC voltage during the cyclohexene ozonolysis experiments to see their effects on the detection of different compounds. Both of these parameters would impact the frequency and/or energy of collisions between ions and gas molecules inside the FIMR, and thus also affect the fragmentation probability. During steady-state cyclohexene ozonolysis conditions in the





chamber, the FIMR pressure was gradually increased from 1.0 mbar to 3.0 mbar with all other instrumental parameters kept constant. The pressure change decreased the overall sensitivity of the instrument, and in Figure 4a we therefore plot the signal intensity of selected compounds normalized to that of $C_6H_{10}H^+$. With the increase of FIMR pressure from 1.0 mbar to 3.0

mbar, we observed relatively higher signals of cyclohexene oxidation products, both monomers and dimers, but lower signals of $C_5H_6H^+$. As discussed in Sect. 3.2, we know that $C_5H_6H^+$ is one of the most abundant fragments of cyclohexene inside the Vocus PTR, and the lower signal at higher FIMR pressure indicates less fragmentation. Therefore, the higher relative signals of oxidation products can be caused by their decreased fragmentation in the instrument. The increase of different species varied significantly as the pressure was increased, with some changing up to three orders of magnitude relative to cyclohexene. This

may indicate differences in their vulnerability to fragment under these FIMR conditions.

Comparatively, FIMR DC voltage had a negligible effect on the ion fragmentation pattern. The DC voltage is placed across the ends of the glass tube of the FIMR to establish the axial electric field in the reactor (Krechmer et al., 2018). In this test, we kept the FIMR pressure constant at 1.4 mbar and gradually decreased the DC voltage from 500 V to 450 V. Since the varied voltages showed no obvious influence on the observations, values lower than 450 V were not probed. As illustrated in Fig. 4b,

the relative signal intensity of different compounds changed very little with varying DC voltage. For cyclohexene dimers, we observed only a slight increase in their relative signal intensity at lower DV voltage, but this may also be caused by a small change in the overall mass-dependent transmission of the system.

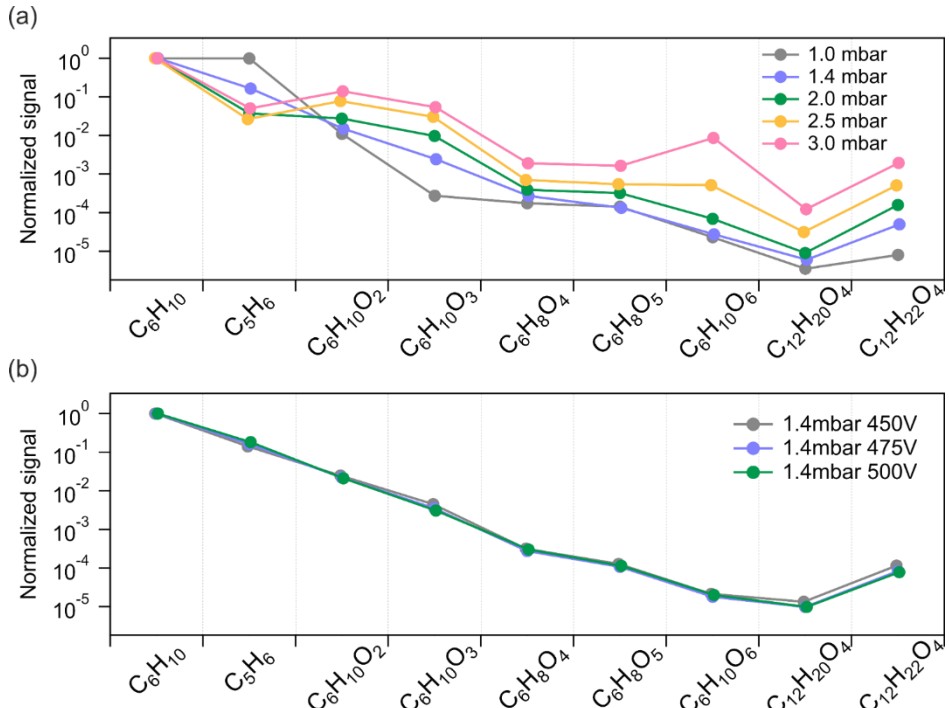

**Figure 4. Influence of instrumental settings on the detection of cyclohexene oxidation products by (a) varying FIMR pressures and**
**(b) varying FIMR DC voltages. The y-axis is the signal intensity of different compounds normalized to that of cyclohexene at each condition.**




### 3.4 Computational results

The experiments showed only very minor signals of potential dimers in the Vocus PTR, and this may be indicative of fragmentation inside the instrument during protonation and ion transport. To assess the role of fragmentation in explaining the

lack of dimer signals, we calculated the survival probabilities of selected ROOR and ROOH species under conditions in the Vocus FIMR. Computations were first performed on the simplest compounds in order to understand the potential fragmentation reactions. The general reaction mechanism found for gas-phase decomposition of protonated peroxides (ROOR' and ROOH) is shown in Figures 5 and 8. Not all of the reaction steps presented are available to each of the studied systems. Except for **R1b**, **R1c**, **R2b** and their subsequent steps, all reaction steps shown in Figures 5 and 8 had already been reported for the

MeOOMe and MeOOH systems by Schalley et al. (1997).

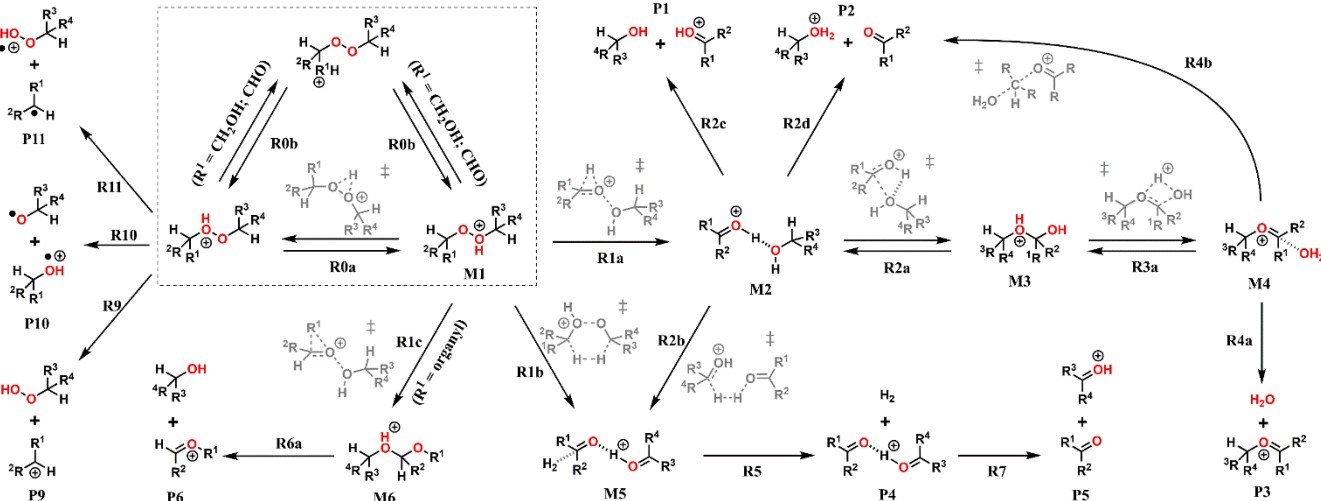

**Figure 5. General mechanism for the decomposition of ROOR following protonation. Species within the dashed box are the initial protonation isomers. Transition states are shown in gray. Some pathways are not available for every ROOR investigated. Compound-specific pathways with yields less than 0.1% for any conditions are omitted for clarity.**

No transition state was found for reaction with $H_3O^+$, so each unique oxygen atom was considered to be a possible protonation site, as long as the reaction is exothermic. This leads to the formation of multiple protonation isomers (dashed box in Figures 5 and 8) if the peroxide is asymmetrical (R≠R') and/or carries oxygenated substituents. Proton transfer between the adjacent peroxyl-oxygen atoms (step **R0a**) in general has very high barriers (~30 and ~25 kcal mol$^{-1}$ for ROOR and ROOH respectively), meaning rates of interconversion between protonation isomers are slow compared to other reaction steps. This

may change if an oxygenated substituent is present, since very low or no reaction barriers were observed from calculations involving proton transfer to and from these groups (steps **R0b**). In these cases, interconversion between different protonation isomers occurs at very short time scales (tautomerism), and they may be considered to be in a Boltzmann distribution. Nevertheless, the possible reaction pathways that were found to be available for each unique protonation isomer were explored.

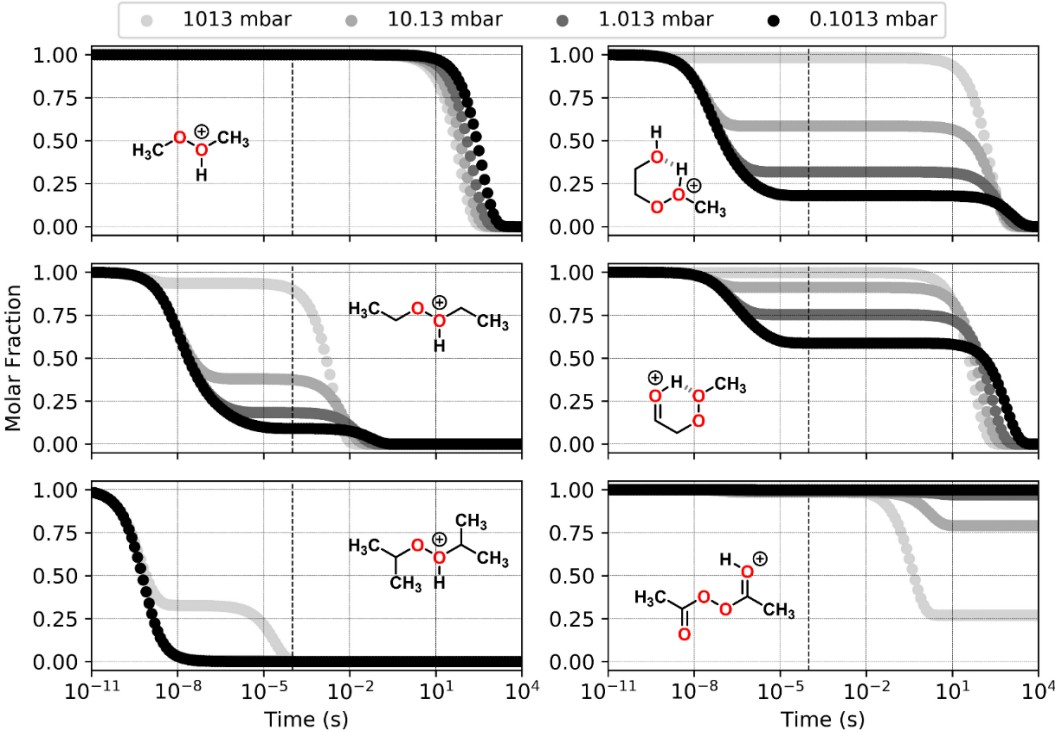

**Figure 6. Reaction dynamics simulation results for model ROOR systems. Graphs represent the time evolution profile of initial protonated reactants due to fragmentation, at 298.15 K. Left-hand side, from top to bottom: MeOOHMe⁺, EtOOHEt⁺ and iPrOOHiPr⁺; Right-hand side, from top to bottom: HOEtOOHMe⁺, HO=EtOOMe⁺ and AcOOAcH⁺(Z). Dashed line indicates 100 μs from the start of the simulation.**

Results from reaction dynamics simulations are shown in Figure 6 and 7 as time-dependent molar fraction profiles of the initial protonated reactant and its (non-fragmentation) isomerization products at different pressures. Most of the shown time profiles present two different stages of reactant molar fraction decay. The earliest stage corresponds to reactions occurring with species carrying excess energy from the preceding (protonation) reaction step, before significant collisional redistribution of energy. With increasing pressure, collisions with bath gas molecules become more frequent, and a smaller proportion of the reactant goes on to form products during this stage. The latter stage corresponds to reactions occurring at thermal rates, when activating collisions with bath gas molecules may be rate limiting, resulting in a positive pressure dependence. These results likely represent a lower limit for (ROOR)H⁺ and (ROOH)H⁺ fragmentation, since the only excess internal energy that was assigned to the initial reactant in these calculations is released during its protonation. In reality, the charged species are generated within a strong electric field, which propels them to supra-thermal speeds. Under this condition, collisions with bath gas molecules have the potential to introduce even more excess internal energy into the reacting system. The following sections will explore the results obtained under this lower limit scenario, while the influence of additional energetic collisions is discussed in the last section.

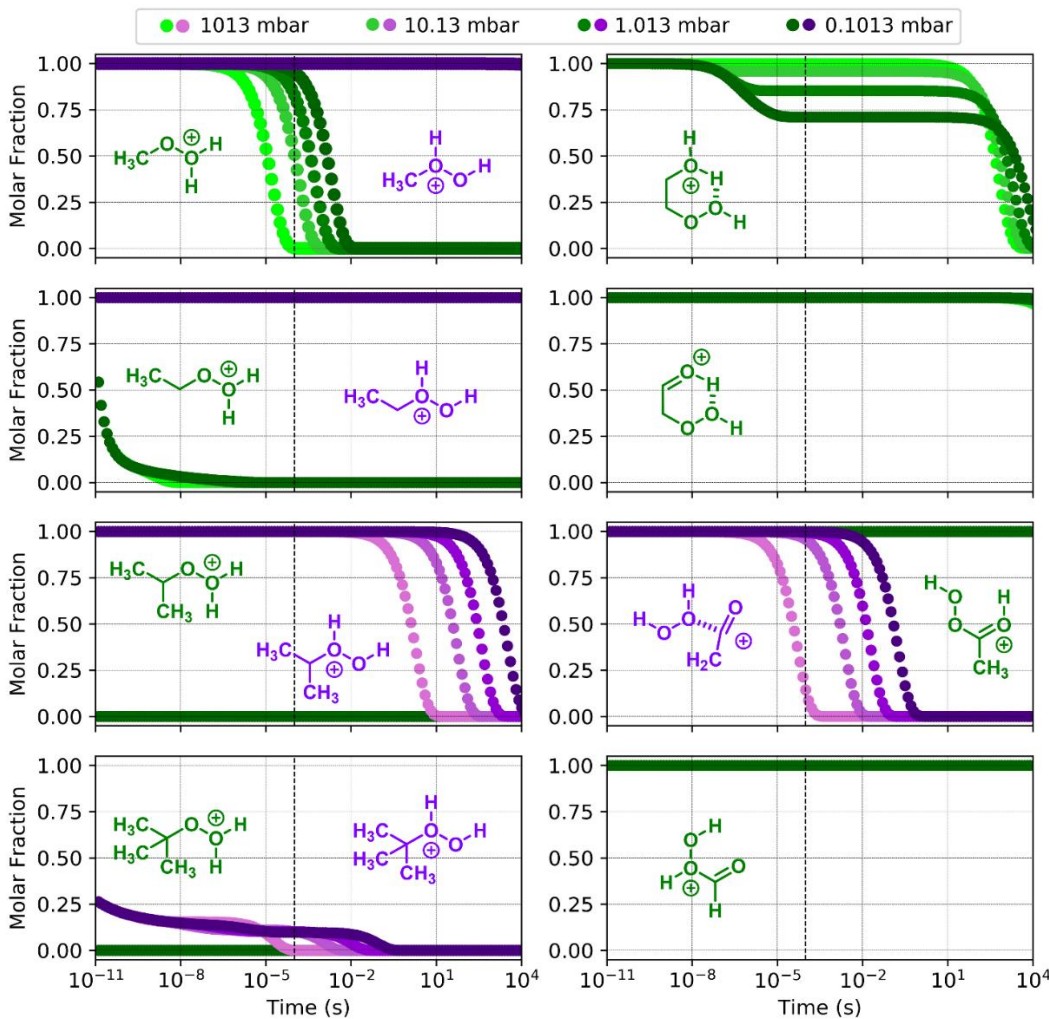

**Figure 7. Reaction dynamics simulation results for model ROOH systems. Graphs represent the time evolution profile of fragmentation of initial protonated reactants, at 298.15 K. Left-hand side, from top to bottom: (MeOOH)H+, (EtOOH)H+, (iPrOOH)H+ and (tButOOH)H+; Right-hand side, from top to bottom: (HOEtOOH)H+, (O=EtOOH)H+, (AcOOH)H+ and (OCHOOH)H+. When two protonation isomers that do not readily interconvert are possible, they are plotted in the same graph with different colors (green and purple). Dashed line indicates 100 μs from the start of the simulation.**

### 3.4.1 Model ROOR' systems

Comparing the results for the alkyl substituted ROOR (Figure 6 and Table 1), faster rates of decomposition (thermal and non-thermal) are observed with increasing degree of substitution (R = Me < R = Et < R = iPr). While >99% of the MeOOMe may survive long enough after protonation to be detected by the mass spectrometer, only ~18% of EtOOEt and virtually none of iPrOOiPr is left after 100 μs, at 1.013 mbar. This is due to a combination of larger exothermicity of protonation by $H_3O^+$ and lower barrier heights for the initial decomposition steps **R1a** and **R1c**, as seen in Figure S6 and Table S1. The first of these factors may be explained by the greater electronic density introduced into the peroxy oxygens by methyl groups via inductive

effects, increasing the proton affinity of the peroxide. The second factor can be understood by invoking hyperconjugative stabilization of the transition states of steps **R1a** and **R1c**. Detailed description of the fragmentation mechanisms and analysis of their energetics is found in the Supplementary Information.

As for the systems containing oxygenated substituents, HOEtOOMe and O=EtOOMe, relatively less fragmentation was observed within relevant time scales. At 1.013 mbar, ~68.2% of HOEtOOMe and ~24.6% of O=EtOOMe are left intact after

100 μs, the major fragmentation product being **P6** in both cases. As described in the beginning of this section, interconversion between the initial protonation isomers was found to be fast. Very low-lying transition states were found for steps **R0b** in the PES of HOEtOOMe, whereas no transition states were found for the O=EtOOMe counterparts. Thus, for the sake of analysis of the rate of decomposition of these two peroxides, all of their initial reaction step transition states can be assumed to be connected to their most stable protonation isomer, HOEtOO**H**Me$^+$ and **H**O=EtOOMe$^+$, shown in Figure S7. The reaction of

protonated AcOOAc, which follows a different mechanism, is shown through its PES in Figure S8. The calculations show that after 100 μs, at 1.013 mbar and 298.15 K, about 18% of protonated AcOOAc is converted to protonated acetic methylcarbonic anhydride (AMCAH$^+$) and ~1% yields fragmentation products AcOOH + Ac$^+$. However, in this case the major product AMCAH$^+$ has the same elemental composition as the reactant, and therefore the mass spectrum would not be considerably impacted. Fragmentation in this system seems to be favored by increasing pressures, as seen from Figure 6, but we emphasize

again that we are now only considering conditions without electric fields causing additional energy to be absorbed through energetic collisions.

With the exception of AcOOAc, the final charged products formed in the decomposition of the model ROOR systems are fragments of the parent molecule, and therefore the signal produced in the mass spectrometer detector would correspond to a different elemental composition. Fragmentation percentage and product distribution for each of these systems are shown in

Table 1.

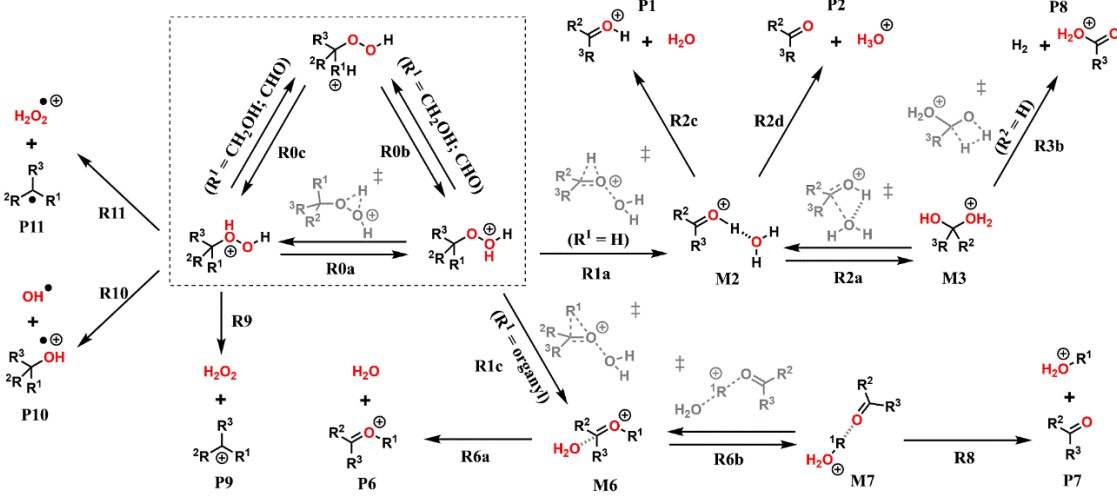

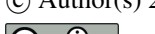



**Figure 8. General mechanism for the decomposition of ROOH following protonation. Species within the dashed box are the initial protonation isomers. Transition states are shown in gray. Some pathways are not available for every ROOH investigated. Compound-specific pathways with yields less than 0.1% are omitted for clarity.**

### 3.4.2 Model ROOH systems

As seen from Figure S9, interconversion between the protonation isomers of alkyl-substituted hydroperoxides (RO**H**OH$^+$ and ROO**H$_2$**$^+$) shows very high reaction barriers (23~26 kcal.mol$^{-1}$) and therefore the two initial reactant species must be considered separately. The ROO**H$_2$**$^+$ protonation isomers showed faster fragmentation rates than their peroxide 'dimer' (ROOR)**H**$^+$ analogues, but both were observed to follow a very similar trend in reactivity. The energy released during protonation by H$_3$O$^+$ is considerably smaller for ROOH systems, which could reduce the proportion of molecules that react at non-thermal rates. On the other hand, the barrier heights for the steps **R1a** and **R1c** are also lower, meaning thermal reaction rates are faster. Given that any of the considered R groups have a greater electron-donating effect than a hydrogen, a ROOH system has a smaller proton affinity but its O-O bond is more polarized compared to a ROOR. Fragmentation percentages and product distributions are shown in Table 1. In contrast, protonation isomer RO**H**OH$^+$ is rather inert. Apart from isomerization into ROO**H$_2$**$^+$, only the three direct dissociation channels (**R9**, **R10** and **R11**) are available to these species. Each of these channels was investigated but, except for **R8** with tButO**H**OH$^+$, none were observed to contribute to fragmentation within significant timescales, under the lower limit of excess energy scenario. At 1.013 mbar, about 89.8% of tButO**H**OH$^+$ fragments into tert-butyl cation + H$_2$O$_2$ within 100 μs, while other RO**H**OH$^+$ remain unreacted in the absence of energetic collisions. Decomposition of the protonated peroxy acids follows a different mechanism, described in detail in the SI. The calculations predict the fragmentation of only ~0.6% of protonated peracetic acid after 100 μs, at 1.013 mbar. No reaction was predicted to occur for protonated performic acid.

**Table 1. Species distribution of each studied ROOR and ROOH obtained from reaction dynamics simulation after 100 μs, at 298.15 K and 1.013 mbar. Results are shown for two scenarios differing in the amount of excess internal energy (E$_{ex}$) assigned to the initial reactant: 1) Only protonation energy (E$_{ex}$ = ΔE$_{zp,prot}$); 2) Protonation energy plus reference energy value from high-speed collisions (E$_{ex}$ = ΔE$_{zp,prot}$ + 0.5 eV). Percentage of unreacted ROOR and ROOH highlighted in bold. $^a$: AMCAH$^+$ is an isomer of the original protonated peroxide. $^b$: End product of channel R1d, shown in the SI.**

| (ROOR)H$^+$ | Species distribution after 100 μs (T = 298.15 K; p = 1.013 mbar) | |
| --- | --- | --- |
| | Low-limit scenario (E$_{ex}$ = ΔE$_{zp,prot}$) | Energetic collision scenario (E$_{ex}$ = ΔE$_{zp,prot}$ + 0.5 eV) |
| Me-OO**H**-Me$^+$ | ROOR (>99.9%) | ROOR (**61.2%**), P1 (35.3%), P2 (3.4%) |
| Et-OO**H**-Et$^+$ | ROOR (**18.3%**), P6 (55.5%), P2 (21.2%), P1 (4.4%), P3 (0.5%) | ROOR (**0.2%**), P6 (80.4%), P2 (15.6%), P1 (3.3%), P3 (0.5%) |
| iPr-OO**H**-iPr$^+$ | ROOR (**~0%**), P6 (55.1%), P2 (32.1%), P1 (12.7%) | ROOR (**~0%**), P6 (58.1%), P2 (30.8%), P1 (11.1%) |
| HOEt-OO**H**-Me$^+$ | ROOR (**31.8%**), P6 (68.2%) | ROOR (**1.4%**), P6 (98.4%), P1 (0.1%), P2 (0.1%) |
| **H**O=Et-OO-Me$^+$ | ROOR (**75.4%**), P6 (24.6%) | ROOR (**4.4%**), P6 (95.6%) |
| Ac-OO-Ac**H**(Z)$^+$ | ROOR (**80.7%**), AMCAH$^+$ (18.4%)$^a$, AcOOH + Ac$^+$ (0.9%) | ROOR (**1.8%**), AMCAH$^+$ (3.8%)$^a$, AcOOH + Ac$^+$ (94.4%) |
| (Cyclohexene + OH) | ROOR (**6.2%**), P10 (92.4%), P9 (1.3%) | ROOR (**~0%**), P10 (95.4%), P9 (4.5%) |
| (Cyclohexene + O$_3$) | ROOR (**0.1%**), P10 (15.4%), P9 (84.3%), P2$^b$ (0.2%) | ROOR (**~0%**), P10 (43.9%), P9 (56.1%) |





| (ROOH)H⁺ | | |
|---|---|---|
| Me-OO**H**-H⁺ | ROOH (**83.0%**), P2 (9.3%), P1 (7.7%) | ROOH (**83.0%**), P2 (9.3%), P1 (7.7%) |
| Me-**H**OO-H⁺ | ROOH (**>99.9%**) | ROOH (**>99.9%**) |
| Et-OO**H**-H⁺ | ROOH (**~0%**), P3 (>99.9%) | ROOH (**~0%**), P6 (>99.9%) |
| Et-**H**OO-H⁺ | ROOH (**>99.9%**) | ROOH (**>99.9%**) |
| iPr-OO**H**-H⁺ | ROOH (**~0%**), P3 (96.8%), P2 (3.2%) | ROOH (**~0%**), P6 (96.6%), P2 (3.4%) |
| iPr-**H**OO-H⁺ | ROOH (**>99.9%**) | ROOH (**38.7%**), P9 (61.3%) |
| tBut-OO**H**-H⁺ | ROOH (**~0%**), P3 (>99.9%) | ROOH (**~0%**), P6 (>99.9%) |
| tBut-**H**OO-H⁺ | ROOH (**10.2%**), P9 (89.8%) | ROOH (**0.1%**), P9 (99.9%) |
| **H₂**OEt-OO-H⁺ | ROOH (**85.3%**), P6 (14.6%) | ROOH (**7.9%**), P6 (84.6%), P1 (6.9%), P2 (0.6%) |
| **H**O=Et-OO-H⁺ | ROOH (**>99.9%**) | ROOH (**74.5%**), P6 (25.5%) |
| **H**Ac-OO-H⁺ | ROOH (**>99.9%**) | ROOH (**>99.9%**) |
| Ac⁺:**H₂**O₂ | ROOH (**99.4%**), Ac⁺ + H₂O₂ (0.6%) | ROOH (**24.8%**), Ac⁺ + H₂O₂ (75.2%) |
| **H**OCH-OOH⁺ | ROOH (**>99.9%**) | ROOH (**>99.9%**) |
| (Cyclohexene + OH) | ROOH (**0.1%**), P9 (99.5%), P6 (0.4%) | ROOH (**~0%**), P9 (99.6%), P6 (0.4%) |
| (Cyclohexene + O₃) | ROOH (**0.1%**), P9 (99.9%) | ROOH (**~0%**), P9 (>99.9%) |

### 3.4.3 Cyclohexene oxidation products

Following the model peroxide molecules, calculations were done to assess whether ROOR and ROOH species produced during
oxidation of cyclohexene are expected to suffer rapid fragmentation following protonation. The investigated peroxides, shown
in Figure 1, are produced from first-generation peroxy radical intermediates of oxidation by OH and $O_3$. Oxidation by OH
radical would be initiated by electrophilic attack on the C-C π bond, with subsequent $O_2$ addition to the radical center carbon,
producing a cyclic β-hydroxy alkylperoxy radical (Aschmann et al., 2012). The formation of the ozonolysis-derived peroxy
radical precursor considered here follows a more complicated mechanism, which is described in detail by e.g. Rissanen et al.
(2014). The hydrocarbon ring is opened during oxidation by $O_3$, resulting in a linear β-oxo alkylperoxy radical.

In all peroxides considered here, the peroxyl group's adjacent C is secondary, and thus the barrier for reaction steps **R1a** and
**R1c** would be expected to be low as observed in iPrOOiPr and iPrOOH systems. On the other hand, these peroxides are also
(at least) β-substituted with O-bearing functionalities, which can be more energetically favorable protonation sites and/or
enable the formation of stabilizing intramolecular H-bonds. As a result, barriers to reaction step **R1a** encountered by these
cyclohexene oxidation products are high, but in general 1~3 kcal.mol⁻¹ lower in comparison to HOEt- and O=Et- substituted
model systems. This value is very similar to the difference in **R1a** barrier height between iPr- and Et- substituted systems.
Concerning **R1c**, the ROOR systems encounter barriers which are 4~5 kcal.mol⁻¹ higher in comparison with model systems,
while the opposite trend is observed for ROOH species. A new channel (**R1d**) was found for the cyclohexene + $O_3$ products,
where the O-O bond scission occurs with concerted 1,6 H-transfer to the far carbonyl-oxygen, shown in Figure S12. The
produced fragments are the same as those resulting from **R1a**, but this step involves a more favorable rearrangement, thus




showing a ~6 kcal.mol$^{-1}$ lower barrier. The PES calculated for the decomposition of these systems is shown in Figures S13-S14.

**Figure 9. Major reaction channels for decomposition of ROOH (left) and ROOR (right) obtained from OH oxidation and ozonolysis of cyclohexene. Branching ratios are shown in gray when relevant.**

Nevertheless, the calculations revealed that the most favorable fragmentation pathways for these systems are **R9** and **R10**, involving direct dissociation of the initial reactant, without any concomitant rearrangement. The energy required for these channels is still high, like the ones observed for the model systems, but it is offset by the much larger energy released during protonation. Results from reaction dynamics simulations (Figure 10) showed that at 1.013 mbar, almost all (99.6%) of the ROOR derived from ozonolysis fragments within ~1 μs following protonation, the major products being **P9** (84.0%) and **P10** (15.4%). About 6.2% of the ROOR derived from OH oxidation survives unreacted after 100 μs, the major products being **P10** (92.4%) and **P9** (1,3%). Virtually all (99.9%) of the ROOH, derived from either ozonolysis or OH oxidation, is predicted to fragment within ~1 μs, leading to the heterolysis products (**P9**). The cationic fragment produced by **R9** suffers rearrangement after dissociation. During geometry optimization, those species spontaneously undergo isomerization into the systems shown in the right-hand side of the ROOH decomposition mechanisms in Figure 9.

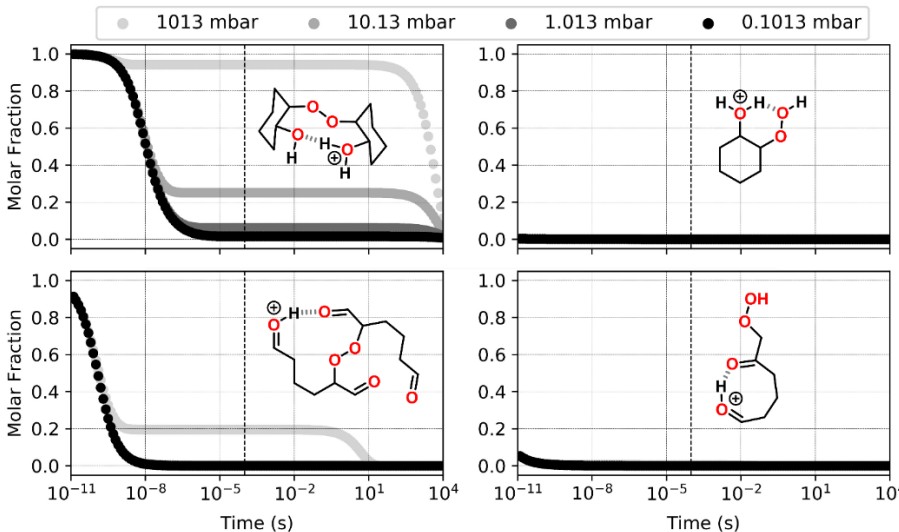





**Figure 10. Reaction dynamics simulation results for the investigated ROOR (left) and ROOH (right) species derived from cyclohexene oxidation by OH radical (top) and O₃ (bottom). Graphs represent the time evolution profile for reaction of initial protonated reactants, at 298.15 K. Dashed line indicates 100 μs from the start of the simulation.**

### 3.4.4 High energy collisions

In order to validate the computational methods, the same set of calculations was done with the decomposition of n-butanol following protonation in the gas phase. In an experiment conducted with this compound in the Vocus PTR, the signal corresponding to $C_4H_9^+$ was observed to be 100 times higher than the signal from the initial charged species $C_4H_{10}OH^+$. The investigated decomposition mechanism is analogous to the one reported for protonated ethanol (Swanton et al., 1991), where the major reaction pathways lead to $H_2O$ or $H_3O^+$ elimination. The calculated PES and reaction dynamics simulation results are shown in Figures S15-S16. The calculations predicted that at 1.013 mbar, only about 48% of n-butanol fragments into 2-butyl cation ($C_4H_9^+$) + $H_2O$ after 100 μs, suggesting that excess energy from protonation alone is not enough to account for fragmentation in the Vocus PTR. High energy collisions provoked by the strong electric field present within the instrument may explain this difference. When assigning an additional 0.5 eV of excess energy to the initial charged reactant on top of the protonation energy, the simulation predicts fragmentation of 98% of protonated n-butanol molecules. This result is much closer to what is observed experimentally, and an energy value of 0.5 eV may serve as a qualitative reference of excess energy derived from supra-thermal speed collisions. With an additional 0.5 eV excess energy, the simulations predict that most of the studied ROOR and ROOH systems fragment significantly within 100 μs, as shown in Table 1. Time evolution profiles of fragmentation under these conditions are shown in Figures S17-18.

### 4 Conclusions

Both computational (Hyttinen et al., 2018) and experimental (Riva et al., 2019) studies have shown that the selectivity of different types of chemical ionization mass spectrometers used in atmospheric chemistry varies considerably. Many of the differences have been attributed to how strongly molecules with different functionalities will bind to the reagent ions, or how favorable a charge transfer from reagent ions to sample molecules are. However, in certain cases this has not been enough to explain observations, and one such example has been the lack of gas-phase dimeric species observable with many PTR instruments, such as the Vocus PTR. In this work, we evaluated the dimer detection both experimentally and through computations.

In the case of α-pinene ozonolysis, a system where e.g. nitrate- and iodide-based CIMS have detected large concentrations of dimers, no verifiable dimer oxidation products were detected with the Vocus PTR-TOF, in agreement with earlier work. A slightly larger, though still marginal, fraction of dimers was observed from the ozonolysis of cyclohexene. The lack of dimers could not be explained by low mass-transmission at the dimer masses, since signals were observed in this range from monomeric oxidation products. The low volatility of any oxygenated dimer species may have further decreased the likelihood



of detection, but given that fragmentation is a common observation in PTR systems, the importance of decomposition reactions remained a viable hypothesis.

With this starting point, we computed the fragmentation pathways and corresponding rates of different peroxide dimers, ROOR. The general trend showed that the smallest ROOR like methyl peroxide showed very little fragmentation following the additional excess energy received upon protonation and the following thermalized collisions, but with increased substitutions and functionalization, more fragmentation was observed. When adding an additional 0.5 eV excess energy to the molecules, as an estimate of the energy absorbed due to energetic ion-molecule collisions driven by the electric fields in the

Vocus ionization region, the vast majority of ROOR dimers were expected to fragment within the 100 μs residence time in this region. The electric fields in the PTR-MS instruments, including the Vocus PTR, are used for guiding and declustering ions, but come with the drawback of enhancing fragmentation reactions for certain species. These results are in good agreement with the observed lack of dimers in the Vocus PTR.

We also performed similar calculations on ROOH hydroperoxides, finding that most of these molecules were also expected to

fragment nearly to 100 % under the conditions of the Vocus PTR. The result is harder to verify experimentally, as the mass spectrometric techniques cannot separate between functional groups within molecules. However, earlier studies have shown that other adduct-forming CIMS types tend to detect more oxygenated species than the Vocus PTR (Riva et al., 2019), lending support to the computational findings of this work. Importantly, our results suggest that the protonation of ROOR and ROOH species does not automatically lead to total fragmentation. However, the added energy from the ion-molecule collisions caused

by electric fields, used for ion guiding or declustering, can add considerably to the fraction of ions undergoing fragmentation. For the Vocus PTR-TOF used in this study, we predict nearly complete fragmentation for almost all studied chemically-labile peroxide species.

PTR-based mass spectrometers are powerful tools for measuring volatile emissions, and newer generation instrument like the Vocus PTR are able to measure also many oxidation products of these emissions. However, the most oxygenated species and

the dimeric species evade detection in most PTR instruments, due to losses caused both by condensation and fragmentation. As such, other CIMS types are likely better suited for the detection of these types of species, though the stability of ions with different functionalities may be worth further study also in these instruments.

**Data availability.** Data are available for scientific purposes upon request to the corresponding authors.


**Author Contributions.** ME and HL conceived the study. HL conducted the chamber experiments. TGA and CDD performed the quantum chemical calculations with the assistance of TK. YL, JZ, WH and CM helped with the measurements. HL and TGA carried out the data analysis. HL, TGA, and ME wrote the paper. All authors commented on the paper.

**Competing interests.** The authors declare that they have no conflict of interest.



**Acknowledgments.** This research has been supported by the H2020 European Research Council (grant nos. COALA (638703), ATM-GTP (742206), and CHAPAs (850614)) and the Academy of Finland (grant nos. 317380, 320094, and 1315600). Haiyan Li acknowledges financial support from the Jenny and Antti Wihuri Foundation. The authors would also like to thank the Knut

and Alice Wallenberg Foundation for financial support (WAF project CLOUDFORM, grant no. 2017.0165).

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
