# Peer review of "Fragmentation inside PTR-based mass spectrometers limits the detection of ROOR and ROOH peroxides"

_Atmospheric Measurement Techniques, 2021_

## Author Response (AR1)

**Author response to referee comments**

We thank the referees for the evaluation of the manuscript and the positive feedback which have greatly helped us improve the manuscript. In the following, we answer the comments point by point (in blue) and mention the changes that we made to our manuscript (in italic) to address the reviewer's concerns and remarks.

**Referee #1**

General comments:

This manuscript describes a combined experimental and theoretical study on the observation of (hydro)peroxides using PTR-based mass spectrometers. The authors argue that due to fragmentation of the peroxides upon addition of the ionizing proton, the (quantifiable) detectability of peroxides is severely hampered, and does not appear to be a viable approach. They clearly document their reasons for this assertion, based on observations of peroxides (or the lack thereof) and an extensive set of theoretical calculations showing facile decomposition pathways for a myriad of protonated peroxides. The paper is well written, reads easily with a clear message founded in high-quality results. I support publication of the manuscript, with only some minor comments below. Note, however, that I am not able to properly assess the experimental part of the paper and focus mostly on the theoretical part.

We thank the reviewer for the positive assessment of our work. We have carefully considered the specific and typographic comments and incorporated them in the revised manuscript. Please find below our detailed response to the reviewer's comments.

Specific comments:

p. 12, line 285: "This may change if an oxygenated substituent is present, since very low or no reaction barriers were observed from calculations involving proton transfer to and from these groups."

Do the authors mean to say that migration to adjacent oxygen groups is slow, but transfer to more distant groups is faster? If so, at what distance would the transfer become sufficiently fast to occur under the conditions in this paper ?

Reply: Yes, the results indicate that proton migration between adjacent oxygen atoms (1,2 shift) are slow, but become very fast when the distance between the O atoms in a molecule increase, as shown for the 1,4 and 1,5 migrations in the HOEt- and O=Et- substituted peroxides. We did not perform a systematic study of the speed of this reaction as a function of the distance of the O-bearing functions, but some insight may be gathered from the information available. In the mechanism of decomposition of performic acid (Fig S11) there is one instance of a 1,3 proton migration between O atoms, for which calculations yielded a high reaction barrier (~26 kcal mol$^{-1}$). However, this transfer does not involve any peroxyl oxygen and the observed barrier may not be representative of migrations occurring in protonated peroxides. Calculations for 1,4 and 1,5 proton transfers either did not find any transition state or yielded very low barriers, thus being very fast reactions. Spontaneous proton transfer to the carbonyl O was observed during geometry optimization of the RHOOH$^+$ and ROOH$_2^+$ species of the cyclohexene ozonolysis product, indicating that 1,7 and 1,8 transfers may also be very fast. Proton migration between O atoms at longer distances were not studied, but we expect lower reaction coefficients to be found with increasing distance, since the specific O…H-O orientation required for the reaction entails a larger entropic penalty. At the shortest distance between O atoms (1,2 transfer), optimal overlap of the orbitals involved in the reaction introduces additional strain into the system, and the transition state of this proton transfer is relatively destabilized. A similar relationship is observed with H-shift reactions occurring with peroxy radicals, as documented by e.g. Vereecken and Nozière 2020.

Vereecken, L. and Nozière, B.: H migration in peroxy radicals under atmospheric conditions, Atmos. Chem. Phys., 20, 7429–7458, 2020, https://doi.org/10.5194/acp-20-7429-2020

The use of molecular dynamics to determine energy transfer parameters is not new, and it might be useful to reference earlier efforts. Examples include

Barker, J. R.: Energy transfer in master equation simulations: A new approach, Int. J. Chem. Kinet., 41, 748–763, https://doi.org/10.1002/kin.20447, 2009.

Barker, J. R. and Weston, R. E., Jr.: Collisional Energy Transfer Probability Densities P(E, J; E′, J′) for Monatomics Colliding with Large Molecules, J. Phys. Chem. A, 114, 10619–10633, https://doi.org/10.1021/jp106443d,

Jasper, A. W., Pelzer, K. M., Miller, J. A., Kamarchik, E., Harding, L. B., and Klippenstein, S. J.: Predictive a priori pressure-dependent kinetics, Science, 346, 1212–1215, https://doi.org/10.1126/science.1260856, 2014.2010.

Reply: A few sentences were added to the main script, in the last paragraph of section 2.2 (Computational methods), citing the suggested references:

*"… While mathematical methods for obtaining these parameters used in master equation methodologies have been developed (Barker 2009; Barker and Weston 2010; Jasper et al. 2014), the complexity of these approaches means that in practice, many researchers use some method to empirically fit the thermal transfer process. In this work, we introduce a simplified method to reliably fit the thermal equilibration profile. In brief, we use classical MD to directly measure the collision driven thermal decay rate, followed by empirically fitting the Lennard-Jones parameters in the stochastic gas collision model used within MESMER to match the collision frequency and the thermal decay measured by MD."*

SI: "In particular, it is apparent that the models used by MESMER significantly underestimate the collision rate between nitrogen gas and a large gas-phase molecule, and therefore when using the default parameters the rate of thermalization is in turn underestimated." and

"The selected parameter values, in special $\sigma$, are much higher than those offered by MESMER as default ($\varepsilon/k_B = 50$ K and $\sigma = 5$ Å)"

I don't understand the emphasis on the default collision model parameters in MESMER. It can only have a single default value, but only by chance would these values be appropriate for any study. Default values should not be used as a black box but rather require explicit deliberation, given that it is well known that collision parameters are dependent on the molecule considered, and extensive work exists determining appropriate values for many compounds experimentally and theoretically. The manual of Multiwell, for example, even has a handy table of collision parameters for a variety of molecular sizes. The values obtained by the authors in the current work are by no means surprising for molecules of the size studied here. I propose the authors remove this awkward consideration that default values in MESMER might somehow have been appropriate, and rather look in collisional energy transfer literature for studies that examine the dependence of the parameters on molecular size to state that the values obtained from the MD calculations are not that unexpected (the references in the Multiwell manual would be good start). There are also methods for estimating such molecule-specific parameters that are not based on quantum chemical or MD calculations.

Reply: We understand that these default parameter values available in MESMER are not suitable for most systems, but a user of the software that is less aware of the theory might make use of them indiscriminately. If one such user reads the study and seeks to replicate the methodology, they are therefore warned to have caution while selecting these parameters. However, we agree that the manner in which this issue was reported in the manuscript

was misleading and could pass the idea that the default values were supposed to be a reasonable universal alternative to be used. We thank the referee for pointing that out. The following pieces of text were changed to improve the message.

The end of section S2.1 was altered to:

*"… Then, we demonstrate how to use our results to obtain the parameters for the stochastic models for thermalization used in MESMER. It is worth noting that the values for these parameters offered by the software as a default should be used with caution, as they may lead to significant underestimation of the thermalization rate of larger molecules."*

The mentioned part of the paragraph in section S2.5 was altered to:

*"The selected parameter values for M1 (EtOOEt) deviate significantly from values empirically fitted for thermalization of hexane ($\varepsilon/k_B$ = 343 K and $\sigma$ = 6.25 Å) (Hippler et al. 1983), a similarly sized molecule, and may represent an unrealistic description of the Lennard-Jones potential well. However, the obtained values were selected to yield a $f_{coll}$ that agrees with the MD results, and do not interfere with any other function in MESMER's model."*

And the cited work was added to the list of references:

Hippler, H.; Troe, J.; Wendelken, H. J. Collisional Deactivation of Vibrationally Highly Excited Polyatomic Molecules. II. Direct Observations for Excited Toluene. J. Chem. Phys. 1983, 78, 6709−6717.

SI: "The overall temperature equilibration must therefore arise from the relatively small asymmetry in this histogram, which we see from Figure S4 becomes more symmetric as the simulation proceeds and the temperatures of the gas and analyte become closer."

I would avoid stating that "the models used by MESMER significantly underestimate the collision rate", as the problem appears to be not so much the models but the inappropriate use of black box defaults. The histograms derived from the MD calculations confirms the collisional models in MESMER, i.e. that the amount of energy transfered is roughly exponential for both up- and down-transitions, and that it is the small asymmetric between these curves that causes thermalization. This has been already described extensively in the literature. Some references to this literature would be appropriate.

Reply: The sentence containing "the models used by MESMER significantly underestimate the collision rate" was removed (see answer to previous comment).

The following reference was added to the sentence quoted in the comment:

*"…which we see from Figure S4 becomes more symmetric as the simulation proceeds and the temperatures of the gas and analyte become closer (Tardy and Rabinovitch, 1977)."*

Tardy, D. C. and Rabinovitch, B. S.: Intermolecular vibrational energy transfer in thermal unimolecular systems, Chemical Reviews, 77, 369-408, 10.1021/cr60307a004, 1977.

The authors do not discuss an E-dependence of delta-Edown. Is this because they do not have sufficient data to derive the dependence, or did they find only a negligible dependence?

Reply: It would be possible and interesting to derive an E-dependence of the delta-Edown parameter, but the objective of the MD simulations in this work was simply to fit this parameter and the two Lennard-Jones parameters to the results for use in the master equation solver.

SI, p. 12, start of lower paragraph: "Reaction dynamics simulations for MeOOH2+ revealed that this species decomposes only at thermal rates".

Add : "across the pressure range considered here"

Reply: Added.

Typographic comments:

Add page numbers to SI

Reply: Added.

p. 6, line 163: 10-5 Hartree : superscript on exponent -5

Reply: Changed.

SI, p. 2, 3th line from bottom: "th eory" ; remove spaces

Reply: Removed.

SI: throughout the text "kcal.mol-1". SI notation of units does not use periods (neither for the abbreviation nor for the multiplication)

Reply: "kcal.mol$^{-1}$" has been changed to "kcal mol$^{-1}$" throughout the text.

SI : "Molecular structures carrying the double-dagger symbol ‡ are transition states." This is evident from the schemes and need not be indicated explicitly.

Reply: Thanks. The sentence has been removed from the figure captions.

**Referee #2**

This manuscript shows the reason why recently developed PTR-based mass spectrometers are not easily able to detect ROOR and ROOH species via extensive experiments and theoretical calculations. The authors used a-pinene and cyclohexene ozonolysis experiments as well as model peroxide, hydroperoxide molecules for the calculation. Among various factors, they have investigated Pressure of FIMR, DC voltage, model compound, and substitutes were investigated. In the experiment, they showed dimers are hardly detected and confirmed that it was not because of low transmission on that mass per charge range. With increasing pressure, less fragmentation was observed and calculated. That was because of more frequent collisions with gas molecules which release excess internal energy. In addition, higher DC voltage in FIMR had a negligible effect on fragmentation. Varying model molecules showed several different results. For example, MeOOMeH+ showed less fragmentation because of their inductive effect and hyperconjugation which stabilize the transition state. Oxygen-based substitutes have a lower level of fragmentation because their initial protonation isomer has a lower barrier for isomerization thus more stable. For hydroperoxides, because of the higher degree of polarizability and smaller proton affinity, they were not easy to be detected by the instrument. This systematic study of model molecules in both experiment and theoretical calculation is well conducted and the manuscript is well written, and the interpretation of the results is reasonable. It is a very informative paper and falls well within the scope of AMT. Therefore, I recommend this paper to publish with some very minor comments and some questions.

We thank the referee for the assessment of our work and the constructive comments. We have revised our manuscript according to the suggestions.

1. Do you have any idea why a-pinene dimers were not detected at all but some cyclohexene dimers are detected?

Reply: According to the chamber experiments, only noisy signals of potential dimers ($C_{18}H_{30}O_2$, $C_{20}H_{30}O_1$, $C_{20}H_{30}O_2$, and $C_{20}H_{32}O_3$) were observed during the ozonolysis of α-pinene. Comparatively, a slightly larger, though still marginal, fraction of dimers ($C_{12}H_{20}O_4$ and $C_{12}H_{22}O_4$) were detected from the ozonolysis of cyclohexene. This difference can be due to the increased functionalization of the α-pinene dimers, as suggested in this study by the quantum chemical calculations. Computational results in Sect. 3.4 show that faster rates of decomposition are expected with increasing degree of substitution and functionalization.

2. I think in the introduction, not much about hydroperoxides is mentioned.

Reply: This is a very valid point. The following sentences were added in the introduction, page 3, line 70.

*"... The compounds above ~300 Th likely contain organic hydroperoxides, ROOH, or organic peroxides, ROOR. The former are mainly formed from the reaction of organic peroxy radicals ($RO_2$) with $HO_2$ or in the process of autoxidation (Crounse et al., 2013), while the latter form from cross reactions of $RO_2$ (Schervish and Donahue, 2020; Tomaz et al., 2021). The CI-APi-TOF instruments using either $NO_3^-$ or $C_4H_{12}N^+$ (butylamine) as reagent ions agreed well for highly oxygenated accretion products, which are believed to be primarily ROOR dimers."*

3. Would it be different if the experiment is conducted in higher relative humidity like 50 %?

Reply: We do not expect that increasing the relative humidity would influence the overall conclusions of this study. Flow tube experiments have shown that the formation of highly oxygenated products from α-pinene oxidation is largely RH-independent (Li et al., 2019). These molecules can mostly be explained by the autoxidation of $RO_2$ following by bimolecular reactions with other $RO_2$ or $HO_2$, rather than from a water-influenced pathway like the formation of a stabilized Criegee intermediate. In addition, one advantage of the Vocus PTR is that the instrument sensitivity shows no dependence on the humidity of the sample air, which is due to the large mixing ratio of water vapor in the FIMR (Krechmer et al., 2018). Therefore, the variations in the relative humidity won't significantly influence the detection capability of the instrument.

Li, X., Chee, S., Hao, J., Abbatt, J. P. D., Jiang, J., and Smith, J. N.: Relative humidity effect on the formation of highly oxidized molecules and new particles during monoterpene oxidation, Atmos. Chem. Phys., 19, 1555–1570, https://doi.org/10.5194/acp-19-1555-2019, 2019.

Krechmer, J., Lopez-Hilfiker, F., Koss, A., Hutterli, M., Stoermer, C., Deming, B., Kimmel, J., Warneke, C., Holzinger, R., Jayne, J., Worsnop, D., Fuhrer, K., Gonin, M., and de Gouw, J.: Evaluation of a New Reagent-Ion Source and Focusing Ion–Molecule Reactor for Use in Proton-Transfer-Reaction Mass Spectrometry, Analytical Chemistry, 90, 12011-12018, 10.1021/acs.analchem.8b02641, 2018.

4. Page 7.

How is the reaction ratio between RO2+RO2 / RO2+HO2?

Do you also have a similar experiment at higher RO2 + HO2? Like in the presence of CO?

Reply: With the Vocus PTR mass spectrometer, while we are able to identify the elemental composition of different compounds, the molecular structures of the compounds remain unknown due to the limitation of the instrument. Therefore, we cannot based on our data determine the exact relative importance between $RO_2+RO_2/RO_2+HO_2$ in this study. However, typically alkene ozonolysis chamber experiments are expected to strongly favor the RO2+RO2 pathway, as also indicated by high yields of ROOR observed with other types of CIMS (Berndt et al., 2015; Zhao et al., 2018; Riva et al., 2019; Schervish and Donahue, 2021).

Unfortunately, a similar experiment in the presence of CO was not performed.

Berndt, T., Richters, S., Kaethner, R., Voigtländer, J., Stratmann, F., Sipilä, M., Kulmala, M., and Herrmann, H.: Gas-Phase Ozonolysis of Cycloalkenes: Formation of Highly Oxidized RO2 Radicals and Their Reactions with NO, NO2, SO2, and Other RO2 Radicals, The Journal of Physical Chemistry A, 119, 10336-10348, 10.1021/acs.jpca.5b07295, 2015.

Zhao, Y., Thornton, J. A., and Pye, H. O. T.: Quantitative constraints on autoxidation and dimer formation from direct probing of monoterpene-derived peroxy radical chemistry, Proceedings of the National Academy of Sciences, 115, 12142-12147, 10.1073/pnas.1812147115, 2018.

Riva, M., Rantala, P., Krechmer, J. E., Peräkylä, O., Zhang, Y., Heikkinen, L., Garmash, O., Yan, C., Kulmala, M., Worsnop, D., and Ehn, M.: Evaluating the performance of five different chemical ionization techniques for detecting gaseous oxygenated organic species, Atmos. Meas. Tech., 12, 2403-2421, 10.5194/amt-12-2403-2019, 2019.

Schervish, M., and Donahue, N. M.: Peroxy radical kinetics and new particle formation, Environmental Science: Atmospheres, 1, 79-92, 10.1039/D0EA00017E, 2021.

5. Page 17, line 383

Can you explain more about why the opposite trend was observed for hydroperoxide?

Reply: We have discussed and came up with a possible explanation for the observed trends. Channel R1c consists in the cleavage of the O-O bond, concerted with migration of an organyl group from the α-OO carbon in the cationic fragment to its adjacent oxygen. Stabilization of the transition state by hyperconjugation is stronger in a system with a secondary α-OO carbon (as in the cyclohexene oxidation derived ROOR and ROOH species), than in a system with a primary α-OO carbon (model HOEt- and O=Et- substituted ROOR and ROOH). This effect alone could explain the lower R1c reaction barrier observed for the cyclohexene derived ROOH when compared to the model compounds. The R groups in the cyclohexene derived peroxides are rather large, and inductive effects raise their proton affinity, as well as the relative stability of the protonated reactants (see Table S3). Two such

large groups are present in ROOR systems, whereas only one is present in ROOH systems. This increased stability of the reactant form of the cyclohexene derived ROOR may raise the reaction barrier enough to overcome the lowering provoked by hyperconjugative stabilization of the TS. In the cyclohexene derived ROOH systems, the greater asymmetry in electron density donation to the peroxyl group may also increase the polarization of the O-O bond, further enhancing its lability.

To address this issue, a small sentence was added to the revised manuscript, page 17, line 395:

*"This may be due to the added stability brought to the reactant by the larger R- groups, which would be stronger in the ROOR systems compared to the ROOH systems."*

6. Supplement S2.4 Page 7. Result and discussion, end of it (with the same page with Figure S4 and Table S1)

"The overall temperature equilibration must therefore arise from the relatively small asymmetry in this histogram, which we see from Figure S4 becomes more symmetric as the simulation proceeds and the temperature of the gas and analyte become closer."

--> It is hard to see how asymmetry (I agree it is small) become more symmetric as simulation proceed. No systematic change with increasing time. Can you give us more explanation about this?

Reply: Figure S4 of the manuscript's SI was altered to include a graph that emphasizes the asymmetry in the histogram and its time evolution. The captions of the figure were accordingly altered:

[Figure]

*"Figure S4: Left: Histogram of the change in the kinetic energy of the colliding gas molecule ΔKE$_{coll}$ during 10 independent simulations with the (EtOOEt)H$^+$ molecule. Right: Cumulative sum of the change in kinetic energy of N$_2$ before/after each collision ($\sum$ΔKE$_{coll}$), multiplied by the number of collisions of that energy. The quantity is multiplied by -1 so that the sum over all collision energies represents the total energy lost by the analyte molecule during each time interval, over the course of 10 individual simulations. The density of N$_2$ is 0.25 atm."*

7. Supplement S2.4 Page 8(Fitting MESMER parameters)

I still did not get why cyclohexene has a longer time for the molecule temperature to be equilibrium as surrounding gas than EtOOHEt+ besides higher collision frequency and larger energy transfer. Can you explain more about this?

Reply: The cyclohexene oxidation product is a large molecule in comparison to EtOOHEt$^+$, so collisions with the bath gas molecules are more frequent, but it also has a lot more vibrational modes and therefore a higher heat capacity. In the simulations, the analyte molecule of each system was assigned the same initial temperature of 800 K. Under this condition, the larger molecule starts the simulation with more internal energy than the smaller one, so even though the rate of collisional energy transfer is faster, thermal equilibrium takes longer to be reached due to the much larger amount of energy it has to transfer to the surrounding gas.

8. This paper can be added as a reference. This provides basic information about the fragmentation patterns from PTR-MS.

Pagonis D, Sekimoto K, de Gouw J. A Library of Proton-Transfer Reactions of H3O+ Ions Used for Trace Gas Detection. J Am Soc Mass Spectrom. 2019 Jul;30(7):1330-1335. DOI: 10.1007/s13361-019-02209-3. Epub 2019 Apr 29. PMID: 31037568.

Reply: Thanks. This reference has been added in the revised manuscript.

Page 3, line 88, "*a process which is known to be common for different types of hydrocarbons in PTR instruments (Tani et al., 2003;Aprea et al., 2007;Gueneron et al., 2015;Pagonis et al., 2019).*"

For correction

1. Supplement page 2, line 6

theory remove empty spaces

Reply: Removed.

2. Add page numbers in supplement

Reply: Page number has been added in the revised supplement.

3. Two S2.4 (Result and discussion, Fitting MESMER parameters)

Reply: "*S2.4 Fitting MESMER parameters*" has been changed to "*S2.5 Fitting MESMER parameters*".

4. Remove dot from kcal.mole-1

Reply: The dot in kcal.mol$^{-1}$ has been removed throughout the text.

5. Supplement Page 11 8th line from the bottom

"stabilised" --> stabilized (unify UK/US English)

Reply: Changed.

[revised manuscript text omitted]

We find that a single exponential function of the form:

(S1)
$$\Delta T(t) = Ae^{-Bt}$$

fits the temperature difference well in all cases we have studied. The inverse of the constant B in Equation S1 can be interpreted as a characteristic time. For the systems we have studied, and with a gas pressure of 1 atm, this time is on the order of 1 to 10 ns; however, there is considerable variation for different molecules. In the examples we focus on here, we note that the cyclohexene derived ROOR' take_s longer to reach the same temperature as the gas compared to the two ethyl peroxide dimers (ca. 5.5 ns vs. 4 ns). By post-processing the trajectories, we obtained

more information about the collision events. We define a collision starting if either a nitrogen atom gets closer than 4 Å to one of the atoms in the analyte molecule, and ending when both nitrogen atoms are further away than 4 Å from all of the atoms in the analyte. One complication is the possibility of multiple gas molecules colliding simultaneously; to mitigate against this rare occurrence we ran more simulations with a reduced gas pressure of 0.25 atm. This analysis allows us to compute the gas collision frequency $f_{coll}$.

[Figure]

**Figure S4:** Left: **Histogram of the change in the kinetic energy of the colliding gas molecule** $\mathit{\Delta KE_{coll}}$ **during** 10̶9̶ **independent simulations with the (EtOOEt)H$^+$ molecule.** Right: Cumulative sum of the change in kinetic energy of $N_2$ before/after each collision ($\sum \mathit{\Delta KE_{coll}}$), multiplied by the number of collisions of that energy. The quantity is multiplied by -1 so that the sum over all collision energies represents the total energy lost by the analyte molecule during each time interval, over the course of 10 individual simulations. **The density of $N_2$ is 0.25 atm.**

[revised manuscript text omitted]